# Harmony Everything!
# Masked Autoencoders for Video Harmonization

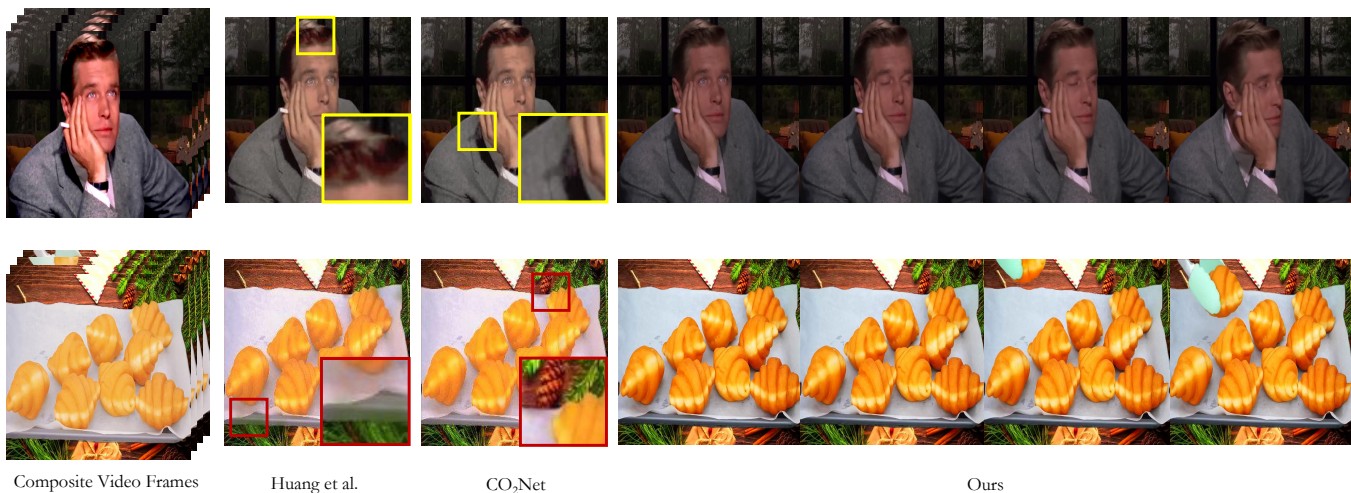

Composite Video Frames (Large-scale Foreground)     Huang et al.     CO$_2$Net     Ours

**Figure 1: The harmonized results from our VHAME and other advanced video harmonization methods under the challenging large-scale foreground setting on our RCVH dataset, demonstrate the effectiveness of our method in complex real scenarios.**

## ABSTRACT

Video harmonization aims to address the discrepancy in color and lighting between foreground and background elements within video compositions, thereby enhancing the innate coherence of composite video content. Nevertheless, existing methods struggle to effectively handle video composite tasks with excessively large-scale foregrounds. In this paper, we propose Video Harmonization Masked Autoencoders (VHMAE), a simple yet powerful end-to-end video harmonization method designed to tackle this challenge once and for all. Unlike other typically MAE-based methods employing random or tube masking strategies, we innovative treat all foregrounds in each frame required for harmonization as prediction regions, which are designated as masked tokens and fed into our network to produce the final refinement video. To this end, the network is optimized to prioritize the harmonization task, proficiently reconstructing the masked region despite the limited background information. Specifically, we introduce the Pattern Alignment Module (PAM) to extract content information from the extensive masked foreground region, aligning the latent semantic features of the masked foreground content with the background context while disregarding the impact of various colors or illumination. Moreover, We propose the Patch Balancing Loss, which effectively mitigates the undesirable grid-like artifacts commonly observed in MAE-based approaches for image generation, thereby ensuring consistency between the predicted foreground and the visible background. Additionally, we introduce a real-composited video harmonization dataset named RCVH, which serves as a valuable benchmark for assessing the efficacy of techniques aimed at video harmonization across different real video sources. Comprehensive experiments demonstrate that our VHMAE outperforms state-of-the-art techniques on both our RCVH and the publicly available HYouTube dataset.

## CCS CONCEPTS

• **Computing methodologies → Appearance representations; Video manipulation; Computer vision**.

## KEYWORDS

Video Harmonization, Video Composite, Masked Autoencoders, Video Harmonization Dataset

*ACM MM, 2024, Melbourne, Australia*

© 2024 Copyright held by the owner/author(s). Publication rights licensed to ACM.
ACM ISBN 978-x-xxxx-xxxx-x/YY/MM
https://doi.org/10.1145/nnnnnnn.nnnnnnn

## 1 INTRODUCTION

The prevalence of fast-paced multimedia platforms like Meta and TikTok has sparked a significant focus on video editing[6], specifically in the fundamental task of video composition, intending to integrate two unrelated videos seamlessly. This involves extracting the content from one video and overlaying it onto another, resulting in a new, cohesive video composition. However, variations in shooting environments or equipment can lead to discrepancies in color

and lighting between the two videos. Consequently, the resulting composite may appear unrealistic when merging contents from these videos. To address this issue, video harmonization was introduced [14], which is to improve the appearance of the foreground (*i.e.,* the region to be harmonized) to seamlessly integrate it with the background (*i.e.,* the target region), achieving a more realistic and delightful composition results.

The applications of video harmonization span various domains, including computer vision tasks, film and television post-production, *etc.,* encompassing video editing [21, 38, 40, 47], video enhancement [45, 46], and virtual production [22, 24]. Existing works on video harmonization can be broadly classified into two categories: *1)* Mapping-based methods [32, 41] primarily employ deep neural networks to directly learn color and feature mappings between input video frames while they often require large amounts of color-labeled training data; *2)* Temporal consistency-based methods [2, 14] leverage spatio-temporal features to maintain natural motion flow across frames, enhancing the visual effects, albeit at the exponential increased computational complexity. Meanwhile, as shown in Figure 1, these methods typically struggle to produce fine-grained videos when the input video contains large-scale inharmonic foregrounds with limited background information, leading to uneven colors or inconsistent foreground details. To this end, our intuition is to propose an end-to-end, simple yet efficient network capable of performing large-scale foreground video harmonization.

In recent years, Mask Autoencoders (MAE) [12] have become prominent in computer vision, particularly with the Masked Image Modeling (MIM) framework, which randomly masks large portions of image patches. MIM allows the encoder to derive latent representations, which are then combined with mask tokens to reconstruct the original input image. Notably, through the masking strategy, MAE operates on a small fraction (*e.g.,* 25%) of the image, enhancing the capacity of the model to learn effectively even from a limited visible area. In image harmonization, LEMaRT[28] introduced the MIM framework to recover the input image, which is composited with randomly masked foreground and ground truth background during pretraining. In the inference stage, an additional fine-tuning process is required to address irregular foregrounds commonly found in real videos. Despite the need for pretraining, it has shown robust competitiveness and performance in the image harmonization task, especially with its more efficient MAE-based model. For more complicated video tasks, VideoMAE [35] explores video content understanding by effectively tackling temporal and spatial redundancy using the masking strategy. However, few methods currently utilize MAE to achieve satisfactory harmonized video results, especially with large foreground areas and limited background information, such a challenged open problem is under-explored in the video harmonization task.

Motivated by the above analysis, we propose Video Harmonization Mask Autoencoders (VHMAE), an end-to-end network competent to large-scale foreground inputs. As shown in Figure 1, we demonstrate the effectiveness of our performance in coping with this large-scale foreground setting, delivering superior results with more natural and realistic colors. In particular, we consider all foreground contents in each frame as the masked area, leveraging both semantic content (*e.g.,* objects pattern) and photometric information (*e.g.,* color and light) to reconstruct harmonized videos. Thus,

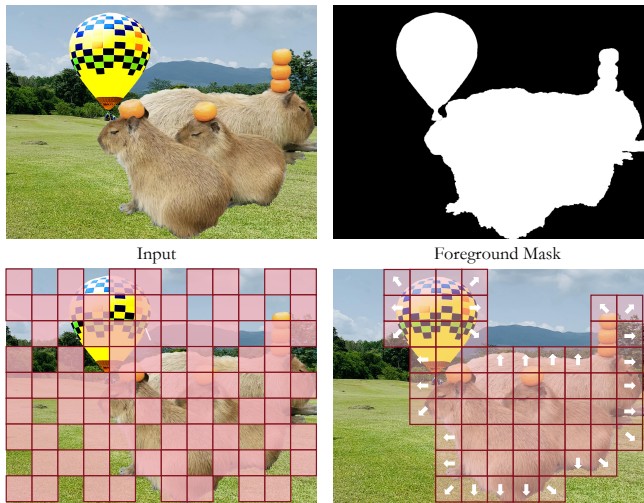

Input             Foreground Mask

Random Masking Strategy     Foreground Masking Strategy (Ours)

**Figure 2: Different masking strategies. Unlike the random or block masking used in the MIM framework, our foreground masking strategy covers all foreground regions. This not only efficiently alleviates the boundary between foreground and background but also allows our model to focus on harmonization while preventing the erroneous acquisition of mismatched color and lighting information, thereby achieving superior harmonization performance.**

our VHMAE incorporates the MIM framework at very high mask ratios (*i.e.,* covering all foreground objects), enhancing model efficiency. Meanwhile, contrary to other MAE-based image/video reconstruction approaches that utilize random or block masking strategies, as illustrated in Figure 2, our method enables the model to directly tackle disharmony areas, achieving refined results.

Our VHMAE consists of two key and novel modules. Firstly, to avoid the model failing to access content information and to mitigate disharmony within the masked area, we introduce the Pattern Alignment Module (PAM), designed to primarily focus on recovering color and light attributes to facilitate harmonization. PAM extracts the semantic information from the masked patches and aligns it with the semantic content of visible patches in the feature space, serving as initial mask tokens for the decoder. Our insight is to enable the decoder to concentrate on recovering the input photometric information, fostering a more coherent and harmonious composition of the foreground (masked) and the background (visible). Secondly, MAE-based methods often result in exhibiting grid-like artifacts, where the output is arranged in a grid pattern corresponding to the split patches, especially in the end-to-end setting. To address this issue, we introduce the Patch Balancing Loss which optimizes the output to minimize gradients in both horizontal and vertical directions, thereby smoothing out the pronounced demarcations in the results. In this manner, our VHMAE excels in extracting vital semantic information from extensive foreground areas requiring harmonization and merges it with the rich photometric information present in the background, leading to

a strong model capable of producing well-harmonized video outputs. Moreover, research in video harmonization is constrained by having only one publicly available synthetic dataset, HYouTube [32]. Given the limitations of synthetic datasets that may not fully reproduce real-world scenarios, we introduce a new dataset for real-composited video harmonization, named RCVH, crafted by meticulously selecting and merging content from various videos, with a deliberate focus on including larger foreground areas enriched with real-world elements. RCVH thus offers a more authentic and demanding challenge for advancements in video harmonization research. Our main contributions can be summarized as follows:

- We propose VHMAE to deal with video harmonization tasks in the practical large-scale foreground setting, which, to the best of our knowledge, is the first end-to-end MAE-based model for video harmonization.
- We devise two key and innovative modules for VHMAE, *i.e.,* Pattern Alignment Module (PAM) for aligning semantic information between foreground and background and preventing disharmony, and Patch Balancing Loss reduces grid-like artifacts in the output caused by split patches.
- We present a new and practical dataset of real-composited video harmonization dataset called RCVH. Extensive experiments on several benchmarks indicate the effectiveness and superior performance of our VHMAE.

## 2 RELATED WORK

### 2.1 Image Harmonization

The purpose of image harmonization is to modify the foreground appearance of a composite image, including its lighting and color, to match the background, thereby creating a visually cohesive scene. Traditional methods [3, 20, 34, 44] typically adjust the foreground color to match the background using low-level color features. Deep learning based methods [1, 5, 9, 10, 15, 17, 31] have played an important role in recent years. Tsai *et al.* [36] proposed the first CNN network for image harmonization by combining semantic segmentation to build a multi-branch network. Hao *et al.* [11] used a self-attention mechanism [39] to propagate relevant features from the background to the foreground. Some methods [4, 25, 43] also focused on high-resolution image harmonization, resulting in better efficiency and higher harmonization performance. Recently, there has been a surge in Transformer-based [8] and diffusion model-based [13] approaches in the field of image harmonization. Liu *et al.* [28] introduced a MAE-based [12] network using the pretraining strategy. They enhanced the Swin Transformer [29] model by integrating both local and global self-attention mechanisms. Such image harmonization methods that are applied directly to video data often produce deteriorated results (like flickering and artifacts) because they fail to account for temporal information.

### 2.2 Video Harmonization

In the video domain, some video processing methods [19, 23] strive to ensure consistency across video frames, but they often entail longer processing times or necessitate additional training modules. Huang *et al.* [14] employed a pixel-by-pixel incongruity discriminator to obtain more realistic harmonization results, and introduced temporal loss to enhance consistency between frames. Ke *et al.* [18] devised a Harmonizer to ensure smooth changes in predicted filter parameters across different frames. Lu *et al.* [32] used color mapping consistency to maintain temporal coherence. However, these methods rely heavily on background information and weaken when handling large foreground areas. In contrast, we propose an innovative MAE-based framework that facilitates an end-to-end, straightforward recovery of all areas requiring harmonization.

### 2.3 Masked Autoencoders

Masked Autoencoders (MAE) [12] excels as a scalable self-supervised learning model in computer vision. It benefits from its lightweight network, which reconstructs the original image using embedded patch features and masked tokens. Inspired by MAE, VideoMAE [35] explores this approach by introducing a large-scale tube masking strategy, mitigating the risk of information leakage from static or minimally moving tokens during reconstruction, tightly tied with temporal correlation. Following this, VideoMAE V2 [37] further enhances performance by introducing a double masking scheme, designed to decrease computational demands and resource consumption. These methods achieve outstanding outcomes in various vision tasks, such as object detection and segmentation, utilizing the pretraining-finetuning framework. However, in the field of video harmonization, to the best of our knowledge, there is currently no established work, especially concerning end-to-end video reconstruction models. Moreover, MAE-based methods often generate grid-like artifacts when reconstructing images or videos directly, making these results unsuitable for immediate use in multimedia applications without additional processing.

## 3 METHODOLOGY

### 3.1 Revisiting Video Masked Autoencoders

MAE [12] utilizes an asymmetric encoder-decoder structure to perform masking and reconstruction tasks on images. VideoMAE [35] extends its application to video, employing tube masking to capture the temporal correlation among frames. Given a $T$ frames input video $V \in \mathbb{R}^{T \times H \times W \times 3}$, it is initially split into regular and non-overlapping patches $P = \{p_i \in \mathbb{R}^{\frac{H}{N} \times \frac{W}{N} \times 3}\}_{i=1}^{T \times N^2}$ (where $N$ is the patch number per row/column in a frame, set to 16 by default), with each patch embedded as tokens. Subsequently, most of these tokens are masked using various masking strategies. The MAE-based methods aim to combine efficiency and high-quality representation learning by reconstructing complete image/video with only the visible tokens, significantly reducing computational demands while capturing deep, semantically rich features. The result is optimized by comparing the predicted masked tokens with the ground truth ones using the Mean Square Error (MSE) loss:

$$\mathcal{L}_{recon} = \frac{1}{T \times N^2} \sum_{p_i \in P} ||p_i - \hat{p}_i||^2, \quad (1)$$

where $\hat{p}_i$ represents the reconstruct masked patches while $p_i$ is the corresponding truth one.

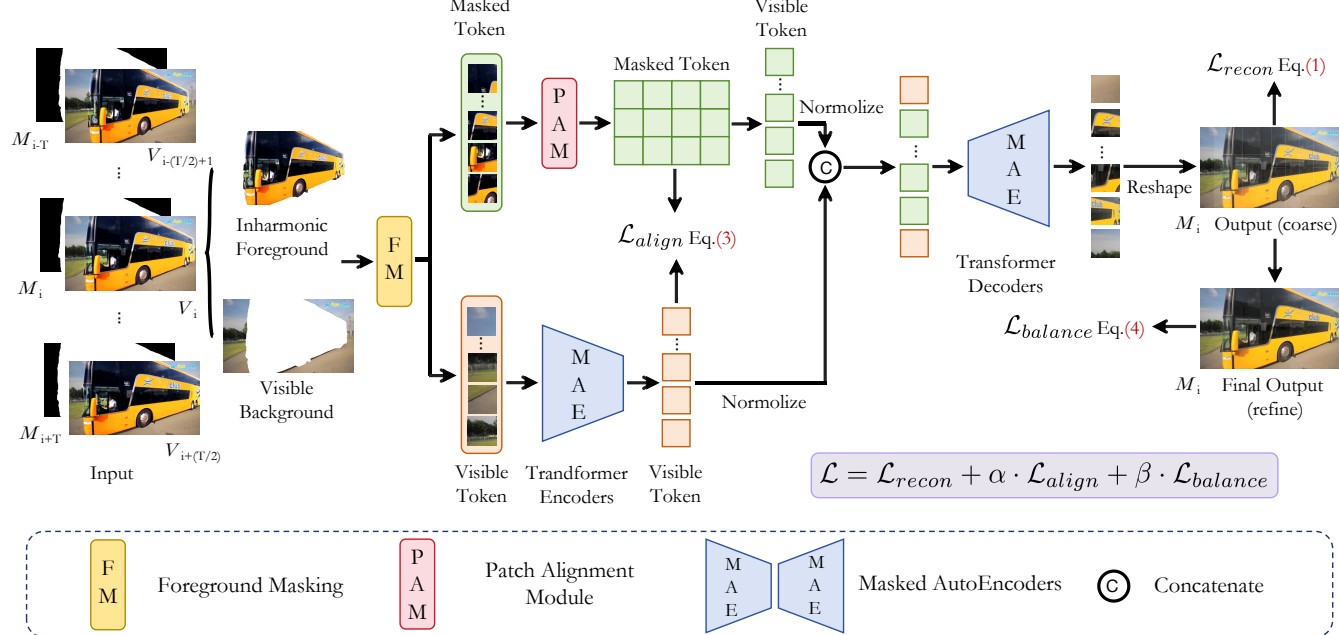

**Figure 3: The architecture of our VHMAE. Given the current composite video frame $V_i$ and its foreground mask $M_i$, our network integrates temporal information from adjacent frames (*i.e.,* from $V_{i-(T/2)+1}$ to $V_{i+(T/2)}$). Firstly, we propose the foreground masking strategy to split the input video frame into the inharmonic foreground and visible background, generating the masked tokens and visible tokens respectively. The masked tokens are then processed by the Pattern Alignment Module (PAM) which aligns their semantic features with those of the visible tokens obtained from the Transformer Encoders, providing the initial masked token with foreground semantic information. This enables the Transformer Decoders to specifically focus on harmonizing color and light, resulting in foreground-background consistency output. At last, we introduce the Patch Balancing Loss ($L_{balance}$) to mitigate grid-like artifacts typical of MAE-based methods, enhancing the final refinement result. We incorporate Reconstruction Loss ($L_{recon}$), Pattern Alignment Loss ($L_{align}$), and $L_{balance}$ to optimize the harmonization process simultaneously.**

### 3.2 Problem Set: Harmonization for Large-scale Foreground Video

In video harmonization, a significant and practical challenge is managing large foreground areas, where the background offers limited benefits to the model, making it difficult to achieve natural and consistent color results.

**Large-scale Foreground Harmonization.** Existing video harmonization methods [14, 18, 32] concentrate on extracting optimal information from the background, However, they often fall short under the large-scale foreground setting, struggling to extract realistic photometric details from the limited background areas. Conversely, our VHMAE leverages the MAE architecture to adeptly reconstruct extensively masked content, seamlessly harmonizing the foreground of each frame by treating it as the masked regions. Our method is inherently well-suited for harmonizing large-scale foreground, effectively extracting rich and meaningful representations, regardless of the large foreground size, even up to 70%.

**Content Information Preservation.** In the Masked Image Modeling (MIM) framework, our setting raises a significant challenge since we mask all foreground regions frame-by-frame, preventing the network from accessing the original foreground data at any time. Our insight is to encourage the network to concentrate on reconstructing the photometric information (like color and light) within tokens for harmonization, rather than being restricted to the content (*i.e.,* the object's pattern) of the video. To alleviate this, we propose the Pattern Alignment Module (PAM) to modify the approach of using randomly initialized mask tokens by imbuing them with preliminary essential semantic information.

### 3.3 VHMAE: Video Harmonization in Maksed Autoencoders

To address the above problem of large-scale foreground for video harmonization, we propose masked autoencoders in masked video modeling, named VHMAE. As shown in Figure 3, we mask all foreground regions in each frame and innovatively design the Pattern Alignment Module (PAM) to steer the network's attention toward the video harmonization task. Additionally, we propose an effective Patch Balancing Loss to refine the MAE-based methodology, targeting the elimination of gird-like artifacts.

**Foreground Masking Strategy.** Distinct from other MAE-based approaches like VideoMAE [35] and MAE-ST [7], our method uniquely leverages all foreground regions as masked targets, rather

than traditional random or block masking strategies. This allows our VHMAE to concentrate on the areas requiring harmonization while preventing the erroneous acquisition of mismatched color and lighting information, thereby achieving superior performance. For the input video, we divide each frame into patches and designate any patch with foreground elements as a masked token. Moreover, this also effectively alleviates the boundary between foreground and background, enhancing their seamless fusion.

**Pattern Alignment Module.** By dynamically masking the foreground across each frame, the model is deprived of semantic information regarding the foreground. To compensate for this, we meticulously devise the Pattern Alignment Module (PAM) to align the pattern features of masked tokens with those derived from visible tokens in the feature space, thereby significantly diminishing the photometric discrepancies between the foreground and background. PAM consists of a sequence of Multi-Layer Perceptions (MLPs) that directly process the foreground, extracting pattern information (*e.g.*, object shape and texture) to form the foreground representation feature $\mathcal{F}_{fore} \in \mathbb{R}^{T \times N_{mask} \times C}$. For background visual tokens, their latent features $\mathcal{F}_{back} \in \mathbb{R}^{T \times N_{vis} \times C}$ can be obtained through the Transformer Encoders in the MAE network, where $N_{mask}$ and $N_{vis}$ are the number of masked and visible tokens respectively, and $C$ represents the embedded feature channel. To this end, we utilize the Gram Matrix [16] to effectively capture and represent the inherent patterns and styles of frames by quantifying the correlations between different features within MLPs, allowing for deep insights into the visual pattern of foreground and background tokens:

$$\mathcal{G} = \sum_{t=0}^{T} \mathcal{F}'_t \cdot \mathcal{F}_t, \quad \mathcal{G} \in \mathbb{R}^{C \times C}, \tag{2}$$

where $\mathcal{F}'_t$ represents the transpose of the feature matrix $\mathcal{F}_t$. Therefore, we optimize the squared Frobenius Norm of the difference between their corresponding Gram Matrices to align the embedded pattern information:

$$\mathcal{L}_{align} = \frac{1}{T \times N_{mask} \times N_{vis}} ||\mathcal{G}_{fore} - \mathcal{G}_{back}||_F^2. \tag{3}$$

**Patch Balancing Loss.** As depicted in Figure 4, a notable challenge encountered in MAE-based image/video reconstruction is the grid-like artifacts, which are characterized by conspicuous, regular grid patterns that overlay the reconstructed image, thereby diminishing its visual quality and fidelity of the output. Such occurrences can be attributed to the reconstruction process, particularly when predicting patch-wise representations for masked regions which tend to align with the grid pattern of the input patches. To mitigate this impact, we propose the Patch Balancing Loss to optimize the pixel gradients in both horizontal and vertical directions across patches, *i.e.*, we aim to minimize the variation between two adjacent pixels in the same direction, effectively eliminating such gird-like disruptive visual inconsistencies. To this end, the Patch Balancing Loss can be depicted as:

$$\mathcal{L}_{balance} = \frac{1}{T} \sum_{t=0}^{T} ||\nabla V_t|| \sim \frac{1}{T} \sum_{x,y,t=0}^{H,W,T} \sqrt{(V_{t,x})^2 + (V_{t,y})^2}, \tag{4}$$

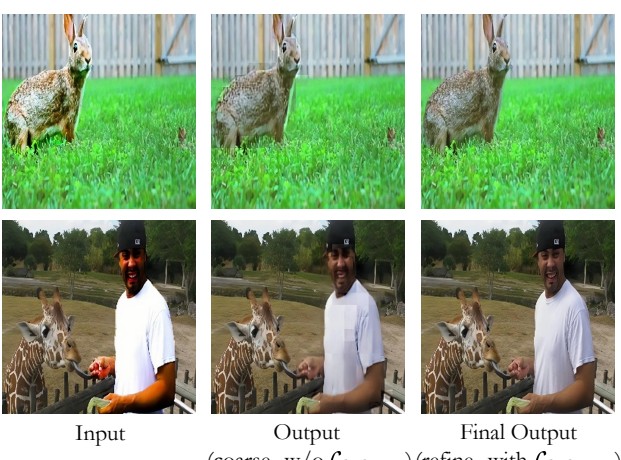

Input      Output      Final Output
(coarse--w/o $\mathcal{L}_{balance}$) (refine--with $\mathcal{L}_{balance}$)

**Figure 4: The grid-like artifacts in MAE-based reconstruction results (middle). We propose the Patch Balancing Loss effectively mitigates this issue, resulting in smoother and more coherent harmonization (right).**

where,

$$V_{t,x}(i,j) = V_{t,i,j} - V_{t,i+1,j},$$
$$V_{t,y}(i,j) = V_{t,i,j} - V_{t,i,j+1}, \tag{5}$$

where, $V_{t,i,j}$ represents the pixel value at the position $(i,j)$ in $t$ frame of the output video.

**Optimization Target.** Our VHMAE incorporates the Pattern Alignment Module (PAM) to facilitate the alignment of foreground and background pattern features, and the Patch Balancing Loss to eliminate grid-like artifacts from the output, eventually resulting in a delicately harmonized video output through the MAE decoder. Consequently, the overall loss function for our optimization can be expressed as:

$$\mathcal{L} = \mathcal{L}_{recon} + \alpha \cdot \mathcal{L}_{align} + \beta \cdot \mathcal{L}_{balance}, \tag{6}$$

where $\alpha$ and $\beta$ are weighting factors utilized to regulate the balance between different components.

### 3.4 RCVH: Real-composited Dataset for Video Harmonization

Real-composited video data, distinguished by its diversity and complexity, poses significant challenges for collection in daily life, primarily due to the absence of corresponding ground truths. Synthetic datasets dominate the field of deep learning-based video harmonization due to their availability through systematic modifications of artificial color lighting. However, this reliance on synthetic data undermines the accurate representation of real-world scenarios. Motivated by this, we meticulously collect a series of real-composited videos, encompassing intricate real-world scenarios, to construct our new dataset, RCVH. This dataset comprises more than 200 raw videos, each spanning $2 - 4$ seconds, sourced from a variety of self-recorded clips and YouTube videos. In particular, we crafted a selection of clear objects to serve as foreground content, segmenting

them from their original videos, and re-integrating with other unrelated video backgrounds. In the end, our RCVH dataset produces 3148 high-quality real-composited video data. Each video presents challenges associated with large foreground areas, underscoring the complexity and diversity of our dataset.

**Composited Data Generation.** RCVH distinguishes from synthetic datasets by eschewing the artificial manipulation of colors and lighting in foreground objects. Instead, it preserves the inherent disparities in appearance between foreground and background elements, thereby ensuring authenticity and realism in the dataset. The data generation consists of two key steps: *1)* Foreground segmentation. We utilize the RVM [26] to identify regions within the video that qualify as potential foreground objects and subsequently segment them to extract individual foreground object frames. *2)* Foreground-background compositing. The foreground object frames, segmented as described earlier, are randomly composited into the background frames of the remaining video, utilizing manual compositing techniques in Adobe Premiere.

**Comparison with Existing Dataset.** Our RCVH significantly enriches the landscape of existing datasets through several distinctive features: *1)* Unlike conventional synthetic datasets [32] that artificially induce discrepancies between foreground and background elements within the same video, every video in RCVH is crafted from real data, with foreground and background elements sourced from different original videos. This method fosters authentic variances in appearance attributable to diverse shooting environments, equipment, and other pertinent factors; *2)* RCVH addresses a broader spectrum of challenges not explored in [32], *i.e.,* handling extensive foreground regions in the real scenarios. *3)* A meticulous manual filtering process is employed to ensure that our dataset meets the highest standards of quality and reliability, guaranteeing its utility for rigorous academic research and practical applications.

## 4 EXPERIMENTS

### 4.1 Experiment Settings

We implement our method using PyTorch and conduct experiments on two NVIDIA A40 GPUs. We set the training batch size to 32 and all models are trained for 100 epochs. Following VideoMAE[35], we employ the AdamW optimizer [30], with an initial learning rate of 0.001, managed by a cosine learning rate scheduler with a weight decay of 0.05. We resize composite frames to $256 \times 256$ during training and testing and apply the same data augmentation (*e.g.,* rotation and flipping), aligning with practices used in $CO_2$Net [32]. Each frame is split into $16 \times 16$ patches, and the weighting factors $\alpha$ and $\beta$ are set to 1.0 by default.

### 4.2 Datasets

We evaluate our method using two distinct datasets, comprising both real and synthetic data.

**HYouTube.** To compare with existing state-of-the-art methods, we use the currently widely used dataset HYouTube [32] on the video harmonization task, which is derived from the large-scale video object segmentation dataset YouTubeVOS [42]. HYouTube comprises 3194 pairs of synthetic composite 20-frame video sequences along with their corresponding ground truths. It includes 2558 video samples for training and 636 samples for testing. Each

video may feature several distinct foreground objects, which are processed independently. The foregrounds in the same video are not allowed to appear in both the training set and the test set.

**RCVH.** In order to better reflect the practicality of the video composition scenario, we present the real-composite dataset, RCVH. This dataset consists of samples derived from two distinct video sources, and importantly, none of the samples have been artificially modified in any way, ensuring authenticity and realistic relevance. Our RCVH dataset contains 3148 high-quality real composites. Since real scenarios lack corresponding ground truths, we use all of our data exclusively for testing. We assess the performance of various methods, including our VHAME, through visual comparisons and user studies, ensuring a thorough evaluation of each method's effectiveness in handling real-world data.

### 4.3 Evaluation Metrics

We evaluate our methods in comparison to others through both qualitative visualization and quantitative numerical metrics. For qualitative analysis, we conduct extensive experiments and select several representative samples for visualization, as illustrated in Figures 5 and 6. In terms of quantitative evaluation, we use metrics including Mean Square Error (MSE), foreground MSE (fMSE), Peak Signal Noise Ratio (PSNR), and foreground Structural Similarity (fSSIM), consistent with $CO_2$Net [32]. Here, fMSE and fSSIM are specifically calculated for the foreground regions only, providing a focused measure of performance where alterations are most critical.

### 4.4 Comparison on Synthetic Dataset

We compare our VHMAE with two categories of advanced methods: *1)* Image harmonization methods include iS$^2$AM [33], RainNet [27], DoveNet [5], and IIH [9]. we treat each video frame as an image and process them separately through these models. *2)* Video harmonization methods contain Huang *et al.* [14], and $CO_2$Net [32], which are specifically designed to address video harmonization challenges.

**Visual Results.** Following $CO_2$Net, we display two adjacent frames from a sample to illustrate our results, as shown in Figure 5. Our method achieves superior temporal consistency compared to the image harmonization method iS$^2$AM, resulting in smoother results between sequential frames. Moreover, when compared to the video harmonization methods, our results more closely align with the ground truth and achieve better harmonized outcomes. This improvement is attributed to our novel foreground-covered masking strategy, which emphasizes the adjacent regions between the foreground and background. Additionally, our method excels in handling large foreground objects (as depicted in the bottom group of Figure 5), producing more realistic colors and lighting, owing to our proposed Prototype Adaptation Module (PAM), which effectively recovers photometric information in the foreground regions.

**Quantitative Results.** As depicted in Table 1, our VHMAE surpasses all current state-of-the-art image and video harmonization methods. It can be found that methods based on color mapping, such as Huang *et al.* and $CO_2$Net, yield slightly inferior results. In contrast, our method operates as an end-to-end model capable of directly predicting and harmonizing the foreground regions through the network, seamlessly integrating the background, thus delivering superior quantitative outcomes.

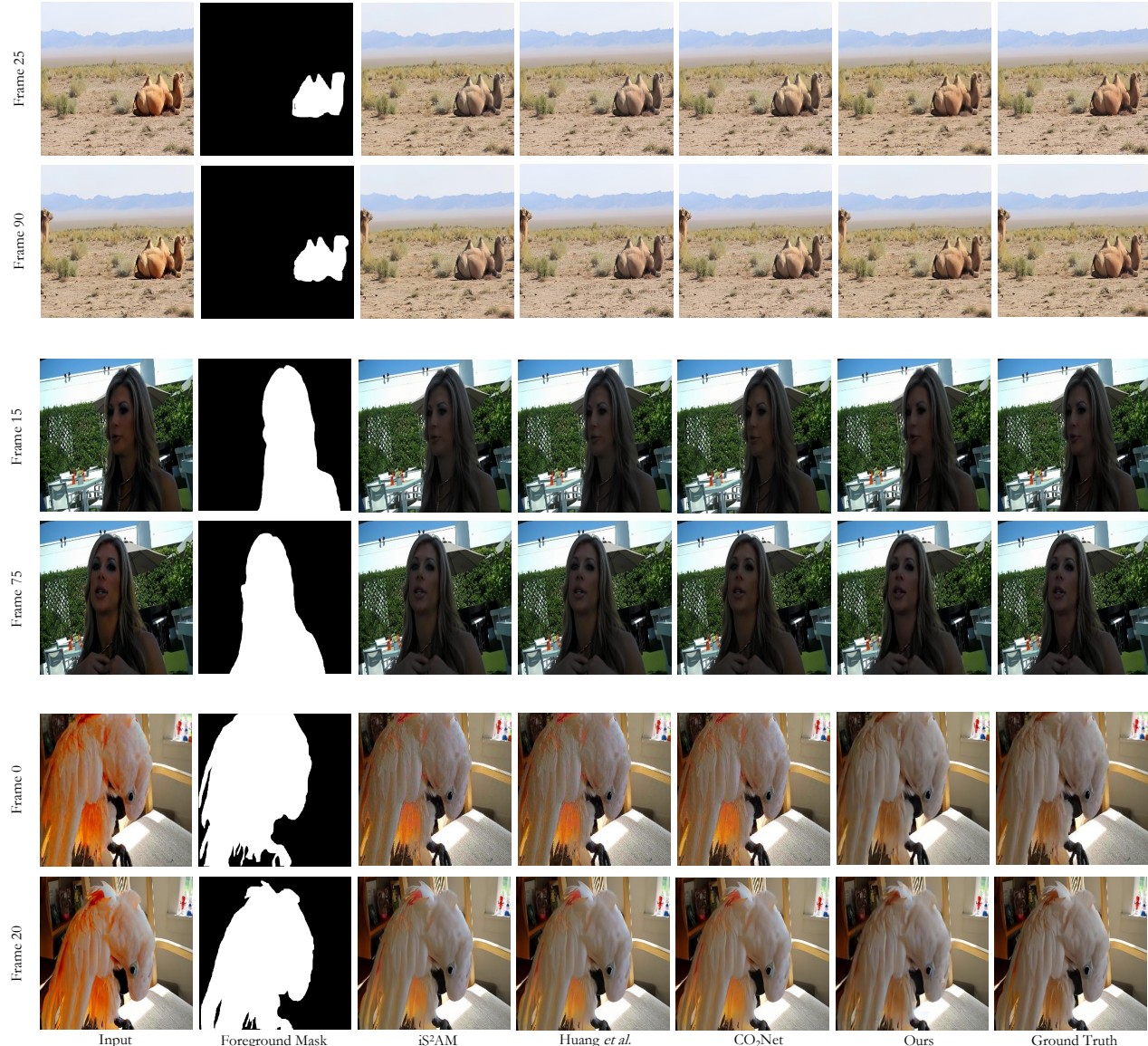

Figure 5: Visual comparison on HYouTube between our network and other state-of-the-art methods.

Table 1: Quantitative comparison between state-of-the-art image and video harmonization methods on the HYouTube dataset.

| Models | Setting | MSE ↓ | fMSE ↓ | PSNR ↑ | fSSIM ↑ |
|---|---|---|---|---|---|
| DoveNet [5] | Image Harmonization | 58.51 | 422.84 | 33.96 | 0.8238 |
| IIH [9] | Image Harmonization | 47.30 | 368.92 | 34.25 | 0.8391 |
| RainNet [27] | Image Harmonization | 49.05 | 374.06 | 34.61 | 0.8338 |
| $iS^2AM$ [33] | Image Harmonization | 28.90 | 203.77 | 37.38 | 0.8817 |
| Huang *et al.* (RainNet) [14] | Video Harmonization | 43.94 | 373.17 | 34.63 | 0.8319 |
| Huang *et al.* ($iS^2AM$) [14] | Video Harmonization | 27.89 | 199.89 | 37.44 | 0.8821 |
| $CO_2Net$ (RainNet) [32] | Video Harmonization | 43.81 | 325.36 | 35.37 | 0.8534 |
| $CO_2Net$ ($iS^2AM$) [32] | Video Harmonization | 26.50 | 186.72 | **37.61** | 0.8827 |
| Our **VHMAE** | Video Harmonization | **25.47** | **173.65** | 37.59 | **0.8832** |

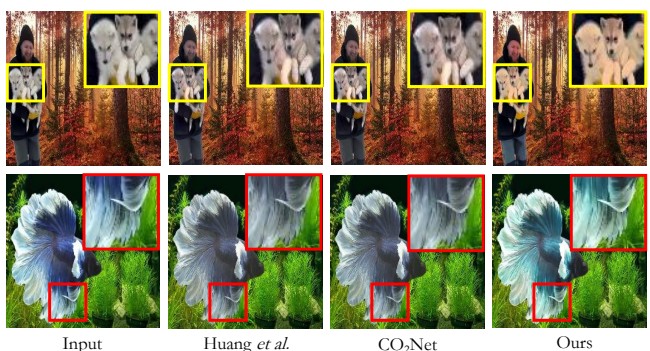

**Figure 6: Visual comparison on RCVH between our network and other state-of-the-art methods.**

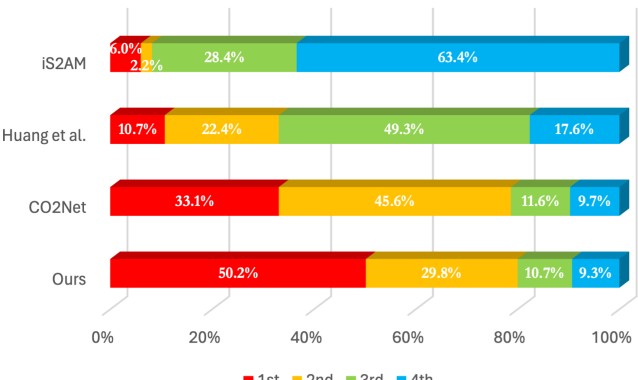

**Figure 7: Rating distribution of the user study.**

## 4.5 Comparison on Real-composited Dataset

We further demonstrate the comparative performance of our VHMAE alongside current advanced video harmonization methods on our proposed real-composited dataset, RCVH. Both our method and other approaches are trained on the synthetic dataset HYouTube and directly tested on the real data from RCVH. This represents a particularly challenging task as it not only evaluates the model's capabilities in video harmonization but also its ability to generalize across different data domains (*i.e.,* synthetic-real). This cross-domain evaluation showcases the robustness and adaptability of the models to handle diverse and realistic video scenarios.

**Visual Results.** As shown in Figure 6, we enlarge the details within the video frames. The results from Huang *et al.* and $CO_2$Net display uneven mottling, where only parts of the foreground blend with the background, indicating color inconsistencies in their harmonization process, whereas our method demonstrates superior harmonized results across the board. We deduce this because their methods suffer from inaccurate color mapping when dealing with new domain data previously unseen by the model, leading to suboptimal results. On the other hand, our end-to-end VHMAE directly avoids these inaccuracies, resulting in natural and coherent frames.

**User Study.** Due to the real-composited data lack of the ground truth, following $CO_2$Net, we conduct a user study to verify the effectiveness of our method. We randomly select 20 real-composited video samples from RCVH and harmonize them using four compared methods (*i.e.,* Huang *et al.* [14], iS$^2$AM [33], $CO_2$Net [32], and ours). We invite 50 participants to attend this user study to test the subjective preference of video harmonization methods. For each video, we play for two seconds, the input data and the four harmonized results will be shown to the participants at the same time without indicating the methods' name. We then ask the participants to rank the quality of the four outcomes from $1^{st}$ (best) to $4^{th}$ (worst) in terms of recovery of brightness, color, and the blend of foreground and background. Figure 7 shows the rating distribution of the user study. Our method receives more "best" ratings, which indicates that our results are more preferred by human subjects.

## 4.6 Ablation Studies

Our method includes two key components: the Pattern Alignment Module (PAM) and the Patch Balancing Loss, which are crucial for achieving desired outcomes. We establish the ablation studies to verify the importance and efficacy of these modules, underscoring their vital contributions to the model's overall performance. As indicated in Table 2, our model performs optimally when two modules are utilized together. This superior performance is attributed to PAM's effective alignment of semantic information between the foreground and background, which leads the model to focus on restoring photometric details for harmonization. Additionally, the Patch Balancing Loss contributes to further smoothing the results, effectively preventing the appearance of grid-like artifacts.

**Table 2: Ablation studies on the Pattern Alignment Module (PAM) and the Patch Balancing Loss (PBL) of our VHAME.**

| Cases | PAM | PBL | MSE ↓ | fMSE ↓ | PSNR ↑ | fSSIM ↑ |
|-------|-----|-----|-------|--------|--------|---------|
| A | | | 26.39 | 188.01 | 36.53 | 0.8821 |
| B | ✓ | | 25.62 | 179.44 | 36.98 | 0.8825 |
| C | | ✓ | 25.98 | 183.15 | 37.34 | 0.8830 |
| **D** | ✓ | ✓ | **25.47** | **173.65** | **37.59** | **0.8832** |

## 5 CONCLUSION

In this paper, we introduce the Video Harmonization Masked Autoencoders (VHMAE), a novel and effective approach that successfully addresses the longstanding challenges of large-scale color and lighting discrepancies in video harmonization. By innovatively treating all foreground regions as masked tokens, our method enhances the integration of foreground elements with their backgrounds, leveraging the contextual information from nearby background regions. Our VHMAE contains two key modules: *1)* the Pattern Alignment Module (PAM), which aligns semantic features across the foreground and background, ensuring a seamless blend regardless of varying colors or lighting conditions. *2)* The Patch Balancing Loss, effectively eliminates common grid-like artifacts, ensuring a visually consistent output. The performance of VHMAE has been empirically validated through extensive testing on our newly proposed RCVH dataset as well as the publicly accessible HYouTube dataset, where it demonstrated superior performance over existing state-of-the-art techniques.

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
