# OpenReview forum: "Harmony Everything! Masked Autoencoders for Video Harmonization"
_acmmm.org/ACMMM/2024/Conference — MM2024 Poster_

### Official Review · Reviewer_6q2R · 2024-05-05

**Rating:** 6
**Confidence:** 4

**Summary:**

This paper is about the large-scale foreground challenge in the video harmonization task. To address this challenge, this paper proposed an MAE-based method named VHMAE that treats all foregrounds in each frame as masked areas, leveraging them to reconstruct harmonized videos.
Then, considering the limitations of existing datasets in large-scale foreground video, this paper constructed a real-composited video harmonization dataset named RCVH.

**Strengths:**

1. The analysis of the large-scale foreground challenge is relatively adequate.
2. The introduction of the masked autoencoder towards large-scale foreground challenge is reasonable.
3. As introduced in the appendix, the source code and the RCVH dataset will be publicly available for research purposes, which are beneficial for the development of video harmonization.

**Limitations:**

1. Is large-scale foreground a common challenge for both image and video harmonization tasks? since according to Sec.2 and Sec. 3.2, the limited background information caused by large-scale foreground is spatial, not specific temporal-wise.
2. It seems that the quantitative experiments are for the image domain since the evaluation metrics (Sec.4.3), including MSE, fMSE, PSNR, fSSIM, are spatial-wise.  The authors should add quantitative experiments for the temporal consistency of video applications.

**Suitability:**

3

---

### Official Review · Reviewer_wohs · 2024-05-24

**Rating:** 3
**Confidence:** 4

**Summary:**

This paper propose Video Harmonization Masked Autoencoders with Pattern Alignment Module and Patch Balancing Loss, which treat all foregrounds as prediction regions and aims to tackle excessively largescale foregrounds of composite videos. Moreover, this work conducts a video harmonization dataset named RCVH.

**Strengths:**

1. This paper builds an effective video harmonization architecture based on VideoMAE, showing promising potential.
2. This work proposes a real-composited video harmonization dataset, which holds value for the community.

**Limitations:**

1. This paper claims to address the issue of large-scale foreground in composite videos, yet it lacks adequate argumentation regarding the challenges. Furthermore, the proposed method fails to incorporate specific designs for this setting, and experimental validation is lacking, including comparisons with other methods.
2. The proposed Patch Balancing Loss lacks innovation and appears to be a common gradient constraint.
3. The experiments are insufficient. First, Table 1 lacks comparisons with the latest methods in image harmonization, especially transformer-based harmonization method, making it difficult to demonstrate its superiority. Second, Table 1 lacks comparisons regarding temporal consistency in video harmonization, which is crucial for evaluating both spatial coherence and temporal continuity in video tasks.

**Suitability:**

2

---

### Official Review · Reviewer_nSK6 · 2024-05-25

**Rating:** 5
**Confidence:** 3

**Summary:**

To enhance the innate coherence and harmony of composite video content, the paper proposes a novel approach called Video Harmonization Masked Autoencoders (VHMAE) to tackle the challenging task of video harmonization, particularly for scenarios with large-scale foregrounds. The authors aim to enhance the coherence of composite video content by addressing discrepancies in color and lighting between foreground and background elements.

**Strengths:**

Novelty in Model: The paper proposes a new model called Video Harmonization Masked Autoencoders ( VHMAE) to treat all foregrounds in each frame as prediction regions and masked tokens are adopted in the training procedure. Such a strategy facilitates both the harmonization task and reconstruction of the masked region effectively, even with limited background information.

The paper proposes  a Pattern Alignment Module (PAM) in the  framework, in order to extract content information from the extensive masked foregrounds to facilitate the harmonization process. This module helps in achieving better results for large-scale foregrounds.

Effectiveness Validation: Experimental comparisons with the existing methods are given to demonstrate the effectiveness of the proposed VHMAE method in complex real scenarios with large-scale foregrounds.

**Limitations:**

The paper lacks of detailed explanations of some components and mechanisms, such as the specific architecture of the VHMAE network and how the PAM module works in practice. More details would strengthen the technical rigor of the paper.

The evaluation is mainly focused on qualitative results and comparisons. More quantitative evaluations, such as objective metrics and ablation studies, would be more convincing to  validate the performance of the proposed method.

The paper does not give discussion on  potential limitations of the proposed method or directions for future work to address those limitations. No new dataset is delivered in the paper.

**Suitability:**

3

---

### Official Review · Reviewer_Haab · 2024-05-28

**Rating:** 3
**Confidence:** 3

**Summary:**

This paper presents an MAE based methods for video harmonization.
Foregrounds went through a designed module called patch alignment module and background was fed into a MAE module.
Features from background and foreground then were fused together and went through a MAE module to produce the final results.
The loss function composed of a reconstruction loss, alignment loss and balance loss.
Experiments were done to compare with SOTA methods and the proposed methods showed better results both qualitatively and quantitatively.

**Strengths:**

Cross-frame MAE is an interesting exploration.

**Limitations:**

The visual results presented were not significant to tell the advantages of the proposed methods (Figure 5).
I am not sure whether the results shown in the Figure 6 are better or worse.

**Suitability:**

3

---

### Meta-Review · Area_Chair_Vkkt · 2024-06-26

**Recommendation:** Accept (Poster)
**Confidence:** 5

**Metareview:**

This paper receives a mixture of reviews, accept, weak accept, and 2 borderline rejects. The reviews of Reviewer Haab are not very informative. The limitations mentioned by Reviewer wohs are well addressed in the rebuttal. The other two reviewers acknowledge the novelty and effectiveness. I agree that this paper proposes a novel approach for video harmonization and the experiments could justify its effectiveness.